Silencing of TRAF5 enhances necroptosis in hepatocellular carcinoma by inhibiting LTBR-mediated NF-κB signaling

Wu Guolin 1 wuguolin28@zju.edu.cn
Wu Fangping 2
Zhou Yang Qing 1
Lu Wenwen 3
Hu Feng Lin 3
Fan Xiaofen 3
1 Department of Traditional Chinese Medicine, Beilun Branch of the First Affiliated Hospital, Zhejiang University School of Medical , Ningbo , China
2 The Second School of Clinical Medicine, Zhejiang Chinese Medical University , Hangzhou , China
3 Department of Traditional Chinese Medicine, The First Affiliated Hospital, Zhejiang University School of Medicine , Hangzhou , China
Xu Zhijie
Electronic publication date: 2023 Jun 22
Publication date: 2023
Volume: 11
Electronic Location ID: e15551
Received 2023 Mar 3; Accepted 2023 May 23
Copyright: © 2023 Wu et al.
Copyright year: 2023
Copyright holder: Wu et al.
License: This is an open access article distributed under the terms of the Creative Commons Attribution License, which permits unrestricted use, distribution, reproduction and adaptation in any medium and for any purpose provided that it is properly attributed. For attribution, the original author(s), title, publication source (PeerJ) and either DOI or URL of the article must be cited.
License URL: https://creativecommons.org/licenses/by/4.0/

Keywords: TRAF5, Hepatocellular carcinoma, Necroptosis, LTBR, NF-κB pathway

Funding: The authors received no funding for this work.

==============================
Background

Hepatocellular carcinoma (HCC) is a common malignancy with poor prognosis and high mortality. This study aimed to explore the oncogenic mechanisms of TRAF5 in HCC and provide a novel therapeutic strategy for HCC.

Methods

Human HCC cell lines (HepG2, HuH7, SMMC-LM3, and Hep3B), normal adult liver epithelial cells (THLE-2), and human embryonic kidney cells (HEK293T) were utilized. Cell transfection was performed for functional investigation. qRT-PCR and western blotting were used to detect mRNA expression of TRAF5, LTBR, and NF-κB and protein expression of TRAF5, p-RIP1(S166)/RIP1, p-MLKL(S345)/MLKL, LTBR, and p-NF-κB/NF-κB. Cell viability, proliferation, migration, and invasion were evaluated using CCK-8, colony formation, wound healing, and Transwell assays. Cell survival, necrosis, and apoptosis were assessed using flow cytometry and Hoechst 33342/PI double staining. Co-immunoprecipitation and immunofluorescence were performed to determine the interaction between TRAF5 and LTBR. A xenograft model was established to validate the role of TRAF5 in HCC.

Results

TRAF5 knockdown inhibited HCC cell viability, colony formation, migration, invasion, and survival but enhanced necroptosis. Additionally, TRAF5 is correlated with LTBR and TRAF5 silencing down-regulated LTBR in HCC cells. LTBR knockdown inhibited HCC cell viability, while LTBR overexpression eliminated the effects of TRAF5 deficiency on inhibiting HCC cell proliferation, migration, invasion, and survival. LTBR overexpression abolished the promotive function of TRAF5 knockdown on cell necroptosis. LTBR overexpression undid the suppressive effect of TRAF5 knockdown on NF-κB signaling in HCC cells. Moreover, TRAF5 knockdown suppressed xenograft tumor growth, inhibited cell proliferation, and promoted tumor cell apoptosis.

Conclusions

TRAF5 deficiency facilitates necroptosis in HCC by suppressing LTBR-mediated NF-κB signaling.

Introduction

Hepatocellular carcinoma (HCC), a common malignancy, is one of the dominant causes of cancer-associated deaths worldwide (Khemlina, Ikeda & Kurzrock, 2017). It has been reported that HCC accounts for approximately 90% of liver cancer, with a prevalence rate of 4.7% of all tumors and a mortality rate of 8.3% worldwide (Yang et al., 2022). Currently, surgical resection, transplantation, and ablation are the standard therapeutic strategies for early-stage HCC (Huang et al., 2022). Some agents, such as sorafenib and lenvatinib, have been used as first-line systemic treatments for advanced HCC (Yano et al., 2022). Despite the development of molecular-targeted curative strategies, the prognosis and outcomes of HCC remain unsatisfactory. Therefore, novel molecular-targeted approaches for HCC treatment are urgently required.

TNF receptor-associated factor 5 (TRAF5) is a core mediator of TNF receptor superfamily members and has significant implications for the differentiation of CD4 T cells and the regulation of various signaling pathways (Nagashima, Ishii & So, 2018). TRAF5 participates in various cellular activities, such as cell proliferation, migration, and apoptosis (Cai & Li, 2019; Willecke et al., 2019; Zhang et al., 2022). Accumulating studies have indicated the carcinogenic role of TRAF5 (Luo et al., 2019; Ma, Duan & Hao, 2020). Thus, reducing the TRAF5 level has been a promising strategy for cancer treatment. For example, the down-regulation of TRAF5 can inhibit the progression of colorectal cancer by suppressing the proliferation, migration, and invasion of cancer cells (Liang et al., 2019). Significantly, TRAF5 silencing has been demonstrated to inhibit the malignant phenotypes of HCC (Ding et al., 2021b). However, the specific mechanisms by which TRAF5 leads to HCC remain to be further elucidated. Of note, TRAF5 can bind directly to the cytoplasmic portion of lymphotoxin beta receptor (LTBR) and interact with LTBR (Piao et al., 2021). LTBR is a type 1 single transmembrane protein and belongs to the tumor necrosis factor receptor (TNFR) family that acts as a switch in the development of secondary lymphoid organs (Piao et al., 2021). Given the significance of LTBR in the formation and regulation of the immune system (Piao et al., 2021), LTBR plays a vital role in tumorigenesis (Bergstrom et al., 2020). Overexpression of LTBR has been reported in some cancers, such as colorectal cancer (Kempski et al., 2020) and neck squamous cell carcinomas (Das et al., 2019), and LTBR has been a potential therapeutic target for cancer.

LTBR can initiate several signaling pathways, especially marked by its role in the activation of the NF-κB pathway (Banach-Orłowska et al., 2019). From the molecular perspective, the ligand binding results in LTBR oligomerization and recruitment of adaptor proteins (TRAFs), ultimately leading to NF-κB activation (Banach-Orłowska et al., 2019). NF-κB is a multifunctional and pivotal dimer transcription factor that modulates various physiological and pathological processes, particularly cell proliferation, inflammation, and the development and progression of cancer (Wu et al., 2018). NF-κB activation usually conduces to the survival of cancer cells by elevating anti-apoptotic genes (Verzella et al., 2020). Accumulating data have shown that aberrant activation of the NF-κB pathway contributes to the onset and progression of tumors, such as colorectal cancer (Soleimani et al., 2020) and breast cancer (Ren et al., 2021). Further, a recent study has indicated the carcinogenic role of NF-κB in HCC, suggesting that inhibition of the NF-κB pathway can prevent the malignant progression of HCC (Dai et al., 2022).

In the past three decades, the effective elimination of tumor cells through the apoptosis mechanism has been a mainstay and target of clinical cancer treatment. Necroptosis is a regulated, caspase-independent cell death form that morphologically resembles necrosis and mechanistically resembles apoptosis (Gong et al., 2019). It is well-recognized that necroptosis is principally mediated by RIP1, RIP3, and MLKL (Gong et al., 2019). The interplay between RIP1 and RIP3 results in RIP3-RIP3 homo-interaction and RIP3 autophosphorylation (Duan et al., 2022). MLKL is recruited and phosphorylated by phosphorylated RIP3 that is then transferred to the cell membrane, leading to necrotizing lesion (Duan et al., 2022). Commonly, RIP1 autophosphorylation at Ser166 (S166) and MLKL phosphorylation at Ser345 (S345) are applied as biomarkers of RIP1 and MLKL activity, respectively (Laurien et al., 2020; Rodriguez et al., 2016). TNF-induced necroptosis is a well-studied pathway of necroptosis. Specifically, the binding of TNF-α to TNFR1 triggers the recruitment of TNF-receptor-associated death domain (TRADD) and its downstream proteins, including RIP1, TRAF2, TRAF5, and the cellular inhibitor of apoptosis (cIAP) 1 and cIAP2 (Abe et al., 2019). TRAF2 and TRAF5 serve as ubiquitin ligases by promoting K-63 poly-ubiquitination in RIP1, contributing to activation of the transforming growth factor B-activated kinase-1 (TAK1)/table 2/3 complex, and ultimately leading to NF-κB activation (Wu et al., 2021). Unlike apoptosis, necroptosis can both accelerate and inhibit tumor growth (Chen et al., 2022). Despite this, necroptosis has been demonstrated to exert an anti-tumor effect in most cases (Tang et al., 2020). Interestingly, a recent study has revealed that promoting Hep3B cell necroptosis via blocking the NF-κB axis can inhibit HCC progression (Bhosale et al., 2022). Nevertheless, little is known about the crosstalk among TRAF5, LTBR, and NF-κB pathway in the necroptosis mechanism of HCC.

Based on the background, we explored the specific oncogenic role of TRAF5 in HCC progression by evaluating the viability, colony formation, migration, invasion, and necroptosis of HCC cells. Additionally, the molecular mechanisms underlying TRAF5 regulating the NF-κB pathway were investigated. This study may uncover the underlying mechanisms by which TRAF5 mediates NF-κB signaling, thus providing a novel therapeutic strategy for HCC.

Materials and Methods

Cell culture

Human HCC cell lines (HepG2, HuH7, SMMC-LM3, and Hep3B), normal adult liver epithelial cells (THLE-2), and human embryonic kidney cells (HEK293T) were obtained from American Type Culture Collection (ATCC, Manassas, VA, USA). All HCC cell lines and THLE-2 cells were cultured in high glucose-Dulbecco’s modified Eagle’s medium (H-DMEM; Gibco, Grand Island, NY, USA) supplemented with 10% fetal bovine serum (FBS; Gibco, Grand Island, NY, USA) at 37 °C with 5% CO2. HepG2, HuH7, and HEK293T cells were also cultivated in H-DMEM that contained 10% FBS and 1% penicillin-streptomycin at 37 °C with 5% CO2 for subsequent functional assays.

Cell transfection and treatments

Two short hairpin RNAs targeting TRAF5 (sh-TRAF5-1 and sh-TRAF5-2) were constructed for the knockdown of TRAF5 expression in HepG2 and HuH7 cells. Besides, two short hairpin RNAs targeting LTBR (sh-LTBR-1 and sh-LTBR-2) were established to down-regulate LTBR expression in HepG2 and HuH7 cells. Overexpression of RNA targeting LTBR (oe-LTBR) was used to up-regulate LTBR expression in HepG2 cells. Scrambled shRNA (sh-NC) and empty vectors were used as controls. GeneChem (Shanghai, China) was used to synthesize all plasmids used in this study. Before transfection, HepG2 and HuH7 cells (2 × 105 cells/ml) were seeded into six-well plates. Lipofectamine 2000 Transfection Reagent (Invitrogen, Carlsbad, CA, USA) was used to transfect plasmids into HepG2 and HuH7 cells and stably transfected cell lines were established. After 48 h of incubation, the cells were collected and quantitative real-time polymerase chain reaction (qRT-PCR) and western blotting were conducted to evaluate transfection efficiency. The primers used in this study are listed in Table 1.

Table 1 Primers used in this study.

Name	Primer sequences (5′ to 3′)	
TRAF5 (Human)	Forward: ATGCCCAGACATCTGTGTCC	
	Reverse: GGGGTCTCTATGCCCAACAA	
LTBR (Human)	Forward: AGCTACTTTCTGACTGCCCG	
	Reverse: TCCTGTGGGGGTGTTTACAG	
NF-κB (Human)	Forward: GGGCAGGAAGAGGAGGTTTC	
	Reverse: TATGGGCCATCTGTTGGCAG	
GAPDH (Human)	Forward: GGAAAGCCTGCCGGTGACTA	
	Reverse: GTGCTAAGCAGTTGGTGGTG	
sh-TRAF5-1 (Human)	Sense strand: GGAAGGTGACAGATTACAA	
	Antisense strand: TTGTAATCTGTCACCTTCC	
sh-TRAF5-2 (Human)	Sense strand: GATGTAGTTACAAAGATAA	
	Antisense strand: TTATCTTTGTAACTACATC	
sh-LTBR-1 (Human)	Sense strand: GCATGAAGATGAAATTATA	
	Antisense strand: TATAATTTCATCTTCATGC	
sh-LTBR-2 (Human)	Sense strand: GCACCTATGTCTCAGCTAA	
	Antisense strand: TTAGCTGAGACATAGGTGC	

HepG2 and HuH7 cells were stably transfected with sh-NC, empty vectors, sh-TRAF5-1, sh-TRAF5-2, sh-LTBR-1, sh-LTBR-2 or oe-LTBR for 48 h. sh-NC- and empty vector-transfected cells served as control cells. To further investigate the mechanisms underlying HCC, HepG2 and HuH7 cells were pre-administered with 12 μM SHN (a necroptosis inducer) for 7 h. To determine the pathway mechanisms of HCC, HepG2 cells were administrated with 1 µM SC75741 (an NF-κB inhibitor) for 24 h.

qRT-PCR

TRIzol reagent (Invitrogen, Carlsbad, CA, USA) was used to extract total RNA from cells according to the manufacturer’s instructions. Then, total RNA was reverse-transcribed into cDNA using an RT-PCR kit (Takara, Beijing, China). The Mx3000P Real-Time PCR System (Stratagene, San Diego, CA, USA) was used for all experiments. The performance was conducted using the following reaction program: 95 °C for 3 min, 40 cycles of 95 °C for 12 s, and 62 °C for 40 s. Gene expression was calculated using the 2−ΔΔCt method. GAPDH served as an internal control. The primers used are listed in Table 1.

Western blotting

Total protein was isolated from HCC cells or tumor tissues after lysis with RIPA lysis buffer (Beyotime, Shanghai, China). The protein concentration was quantified using a BCA kit (Beyotime, Shanghai, China). The obtained proteins were separated using 10% sodium dodecyl sulfate-polyacrylamide gel electrophoresis (SDS-PAGE; Beyotime, Shanghai, China), followed by transfer onto polyvinylidene difluoride membranes (PVDF; Beyotime, Shanghai, China). Subsequently, membranes were sealed with 5% nonfat milk (Beyotime, Shanghai, China) for 1 h. Afterward, the membranes were incubated with primary antibodies at 4 °C overnight, and subsequently with goat anti-rabbit IgG H&L (HRP) (1:10,000, #ab205718; Abcam, Cambridge, UK) secondary antibody for 1 h. Then, the membranes were visualized using enhanced chemiluminescence (ECL) kit (Pierce, Rockford, IL, USA), and protein bands were captured using a Tanon-3500 Image Analyzer (Tanon, Shanghai, China). The detailed information of primary antibodies used was as follows: anti-TRAF5 (1:1,000, #ab137763; Cell Signaling Technology, Danvers, MA, USA), anti-RIP1 (1:1,000, #ARG40183; Arigo, Taiwan, China), anti-p-RIP1 (Ser166) (1:1,000, #ARG66476; Arigo, Taiwan, China), anti-MLKL (1:1,000, #ab184718; Abcam, Cambridge, UK), anti-p-MLKL (Ser345) (1:1,000, #ab196436; Abcam, Cambridge, UK), anti-LTBR (1:500, #ab70063; Abcam, Cambridge, UK), anti-NF-κB (1:1,000, #ab32536; Abcam, Cambridge, UK), anti-p-NF-κB (1:1,000, #ab76302; Abcam, Cambridge, UK), and anti-GAPDH (1:2,500, #ab9485; Abcam, Cambridge, UK). The protein expression was normalized using GAPDH as the internal control.

Cell counting kit-8 (CCK-8) assay

CCK-8 assay was performed to determine the viability of HepG2 and HuH7 cells. According to the supplier’s instructions, cells were seeded into 96-well plates (100 μL cell suspension/well) and subsequently treated with specific reagents for 24, 48, and 72 h at 37 °C in a 5% CO2 atmosphere. Thereafter, 10 µL of CCK-8 solution was added to each well, and the cells were incubated for another 2 h. Optical density (OD) was measured at 450 nm using a microplate reader (DR-3518G; Hiwell Diatek, Wuxi, China).

Colony formation assay

HepG2 and HuH7 cells (200 cells/well) were seeded in six-well plates and cultured for 14 days at 37 °C with 5% CO2. After rinsing twice with phosphate-buffered saline (PBS), the cells were fixed in methanol (1 mL) for 15 min and stained with 0.5% crystal violet (Beyotime, Shanghai, China) for 20 min. The plates were dried at 25 °C, and images were photographed under a microscope (CKX53; Olympus, Tokyo, Japan).

Cell wound healing assay

Cell wound healing assay was performed to detect the migratory ability of HepG2 and HuH7 cells. HepG2 and HuH7 cells (3 × 105 cells/well) were plated in six-well plates and then cultivated at 37 °C overnight. Thereafter, the cell monolayer was scratched using a sterilized 200 pipette tip. Subsequently, the cells were incubated in serum-free DMEM for 24 h at 37 °C and 5% CO2. Images of the wounds were captured under a microscope (CKX53; Olympus, Tokyo, Japan).

Cell invasion assay

Transwell chambers were used to evaluate the invasive ability of HepG2 and HuH7 cells. Cells were seeded in H-DMEM with 10% FBS at a density of 1 × 105 cells/mL, and the upper chamber was pre-coated with 20% Matrigel. The lower chamber contained 600 μL H-DMEM supplemented with 20% FBS. Next, 200 μL of the cell suspension was added to the lower chamber. After 24 h of incubation, the cells were washed with PBS, fixed with methanol at 4 °C for 30 min, and stained with 0.1% crystal violet for 20 min. Images of 12 random fields were captured using a microscope (CKX53; Olympus, Tokyo, Japan). ImageJ software (v1.8.0; National Institutes of Health, Bethesda, MD, USA) was used to quantify cell invasion.

Flow cytometry

Flow cytometry was performed to assess cell survival, necrosis, and apoptosis. After washing twice with PBS, cells were reconstituted in 300 µL binding buffer and stained with 5 μL Annexin V-fluorescein isothiocyanate (FITC; Beyotime, Shanghai, China) for 15 min and 10 µL propidium iodide (PI; Beyotime, Shanghai, China) for another 10 min at 25 °C in the dark. A flow cytometer (CytoFLEX S; Beckman, Miami, FL, USA) was used to measure the survival, necrotic, and apoptotic ratios using Cell Quest software (BD Biosciences, Franklin Lakes, NJ, USA).

Hoechst 33342/PI double-staining

A Hoechst 33342/PI double-staining kit (Solarbio, Beijing, China) was used to evaluate cell survival, necrosis, and apoptosis according to the manufacturer’s instructions. HepG2 and HuH7 cells were seeded into six-well plates (5 × 105 cells/well) and cultivated at 37 °C with 5% CO2 overnight. After washing twice with PBS, the cells were stained with Hoechst 33342 for 10 min and PI for another 10 min. After 30 min of incubation, the cells were washed with PBS, and images were captured under a microscope (CKX53; Olympus, Tokyo, Japan).

Co-immunoprecipitation (co-IP)

HEK293T cells were lysed using Pierce IP Lysis Buffer (Thermo Fisher, Rochester, NY, USA) according to the manufacturer’s instructions. Then, the cell lysates were incubated with anti-TRAF5 (1:1,000, #ab137763; Abcam, Cambridge, UK), anti-LTBR (1:500, #ab70063; Abcam, Cambridge, UK), anti-IgG (negative control, 1: 1:10,000, #ab109489; Abcam, Cambridge, UK), and Dynabeads™ Protein G (Thermo Fisher, Rochester, NY, USA) at 25 °C overnight with shaking. Thereafter, the beads were washed and centrifuged (1,000 g for 1 min; 4 °C). After boiling denaturation with loading buffer, the protein precipitate was obtained and used for western blot analysis.

Immunofluorescence

HEK293T cells were fixed with 4% paraformaldehyde for 15 min, permeabilized with 1% Triton X-100 for 10 min, and blocked with 3% BSA (Beyotime, Shanghai, China) for 30 min. Cells were then incubated with anti-TRAF5 (1:100, #ab137763; Abcam, Cambridge, UK) and anti-LTBR (1:500, #ab70063; Abcam, Cambridge, UK) primary antibodies. After washing three times with PBS, the cells were incubated with goat anti-rabbit IgG H&L (Alexa Fluor® 488) secondary antibody (1:10,000, #ab150077; Abcam, Cambridge, UK), and the nuclei were stained with DAPI. Fluorescence images were captured using a confocal imaging system (UltraVIEW VoX; Perkin Elmer, Waltham, MA, USA).

Animal model and treatments

Twelve pathogen-free BALB/c nude mice used in this study (age, 4–5 weeks; weight, 22 ± 2 g) were purchased from GemPharmatech Co. Ltd., Nanjing, China. All mice were housed under a 12-h light/dark cycle at 22 ± 3 °C with the humidity of 35–55% and allowed ad libitum access to food and water. All animal experiments were approved by the Animal Ethics Committee of Xiamen University (No. XMULAC20220034-4), and complied with the National Institutes of Health Guidelines for the Care and Use of Laboratory Animals.

BALB/c nude mice were randomly classified into the sh-NC and sh-TRAF5 groups (n = 6). The sh-NC- or sh-TRAF5-transfected HepG2 cell suspension (1 × 106 cells/100 μL) was subcutaneously injected into the right dorsal flank of BALB/c nude mice in the corresponding group. Therefore, a xenograft model was established in this study. Tumor growth was monitored every day, and tumor volume (mm3, calculated as length × width2/2) was measured every three days. After four weeks of injection, all mice were euthanized by intraperitoneal injection of 150 mg/kg pentobarbital. Then, tumor weight was measured, and tumor tissues were collected for subsequent experiments.

Immunohistochemistry

Xenograft tumor tissues from mice were paraffin-embedded and sliced at 4 μm. Tumor sections were dewaxed with xylene for 20 min, followed by graded concentrations of ethanol (100%, 95%, 90%, 80%, and 70%) for 5 min at 25 °C. Antigens were obtained by boiling in sodium citrate solution (pH 6.0) for 2 min, and endogenous peroxidase was blocked by soaking in 3% H2O2 for 20 min at 25 °C. The sections were incubated with 50 μL of normal goat serum for 15 min at 25 °C without washing. Thereafter, 50 μL of anti-Ki67 (1:200, #ab15580; Abcam, Cambridge, UK) was added to the sections, which were then incubated at 4 °C overnight and washed three times with PBS. Subsequently, the sections were incubated with a goat anti-rabbit IgG H&L (Alexa Fluor® 488) secondary antibody (1:10,000, #ab150077; Abcam, Cambridge, UK) for 15 min at 25 °C. After development with 3, 3′-diaminobenzidine (DAB; Beyotime, Shanghai, China) at 25 °C, the sections were observed under a microscope (CKX53; Olympus, Tokyo, Japan). The sections were then counterstained with hematoxylin for 30 min, rinsed with running water for 10 min, and dehydrated with xylene and ethanol. Finally, images were captured using the microscope.

Terminal deoxynucleotidyl transferase-mediated dUTP nick-end labeling (TUNEL) staining

TUNEL kits (Beyotime, Shanghai, China) were used to measure cell apoptosis in tumor tissues in accordance with the manufacturer’s instructions. Prepared paraffin-embedded sections were dewaxed and dehydrated in graded concentrations of ethanol and then incubated with 50 μL of proteinase K solution at 37 °C for 30 min. Thereafter, the sections were incubated with the reaction solution for 1 h at 37 °C, washed thrice with PBS, and stained with DAPI for 10 min at 25 °C in the dark. Finally, the sections were photographed using a confocal imaging system (UltraVIEW VoX; Perkin Elmer, Waltham, MA, USA).

Statistical analysis

All data were presented as mean ± standard deviation. Every experiment was conducted thrice. GraphPad 7.0 software (La Jolla, CA, USA) was used for the statistical analysis. Student’s t-test or one-way ANOVA was performed to compare differences between groups. P < 0.05 was considered statistically significant.

Results

TRAF5 knockdown restrains the malignant progression of HCC cells

TRAF5 contributes to the progression of HCC, as evidenced by previous studies (Ding et al., 2021b; Jiang et al., 2020). Thus, we initially examined TRAF5 expression in THLE-2 and HCC cell lines HepG2, HuH7, SMMC-LM3, and Hep3B. and confirmed the up-regulated expression of TRAF5 in these HCC cell lines when compared to THLE-2 cells (P < 0.05; Figs. 1A and 1B). Notably, TRAF5 expression levels in HepG2 and HuH7 cells were markedly higher than those in SMMC-LM3 and Hep3B cells (P < 0.05; Figs. 1A and 1B). Therefore, HepG2 and HuH7 cells were chosen for subsequent experiments.

Figure 1 TRAF5 was overexpressed in HCC cells.

(A) Detection of TRAF5 mRNA expression in HCC cell lines (HepG2, HuH7, SMMC-LM3, and Hep3B) and normal adult liver epithelial cells (THLE-2) using qRT-PCR. (B) Detection of TRAF5 protein expression in HepG2, HuH7, SMMC-LM3, Hep3B, and THLE-2 cells using western blotting. Data were expressed as mean ± standard deviation. *P < 0.05 and **P < 0.01 vs THLE-2 group; #P < 0.05 and ##P < 0.01 vs HepG2 group; &&P < 0.01 vs HuH7 group. TRAF5, TNF receptor-associated factor 5; HCC, hepatocellular carcinoma; qRT-PCR, quantitative real-time polymerase chain reaction.

HepG2 and HuH7 cells were stably transfected with sh-TRAF5-1 and sh-TRAF5-2 and their transfection efficiency was evaluated using qRT-PCR to determine the appropriate sh-TRAF5 for subsequent functional assays. Our results showed that TRAF5 mRNA expression in both HepG2 and HuH7 cells were significantly reduced following sh-TRAF5-1 and sh-TRAF5-2 transfection compared with empty vector transfection (P < 0.01; Fig. S1A). Meanwhile, sh-TRAF5-1 and sh-TRAF5-2 markedly inhibited the viability of HepG2 and HuH7 cells compared with empty vectors (P < 0.01; Fig. S1B). Besides, we did not find significant differences in TRAF5 mRNA expression and cell viability between sh-TRAF5-1- and sh-TRAF5-2- transfected HepG2 and HuH7 cells (P > 0.05; Figs. S1A and S1B), and we selected sh-TRAF5-1 for the following assays.

To further confirm the oncogenic role of TRAF5 in HCC, HepG2 and HuH7 cells were stably transfected with sh-NC (control) or sh-TRAF5. We observed that sh-TRAF5 notably down-regulated TRAF5 expression in HepG2 (Figs. 2A and 2B) and HuH7 cells (P < 0.01; Figs. S2A and S2B), indicating the successful inhibition of TRAF5 in HCC cells by shRNA. Functional assays showed that sh-TRAF5 markedly suppressed the viability (P < 0.05 at 72 h), colony formation (P < 0.01), migration (P < 0.01), and invasion (P < 0.01) of HepG2 cells (Figs. 2C–2F) and HuH7 cells (P < 0.01; Figs. S2C–S2F).

Figure 2 TRAF5 knockdown suppresses the malignant progression of HCC cells.

(A and B) Detection of TRAF5 expression in HepG2 cells using qRT-PCR and western blotting. (C) Detection of the viability of HepG2 cells using CCK-8. (D) Detection of the colony formation of HepG2 cells. (E) Detection of the migration of HepG2 by cell wound healing assay (Scale bar = 50 μm). (F) Detection of the invasion of HepG2 cells using Transwell assay (Scale bar = 50 μm). (G) Detection of the protein expression of p-RIP1 (S166)/RIP1 and p-MLKL (S345)/MLKL in HepG2 cells using western blotting. (H) Detection of the survival, apoptotic, and necrosis ratios of HepG2 cells using flow cytometry. (I) Detection of the necrosis and apoptosis of HepG2 cells using Hoechst 33342/PI double-staining (Scale bar = 50 μm). Data were expressed as mean ± standard deviation. *P < 0.05 and **P < 0.01 vs sh-NC group. TRAF5, TNF receptor-associated factor 5; HCC, hepatocellular carcinoma; qRT-PCR, quantitative real-time polymerase chain reaction; CCK-8, cell counting kit-8; RIP1 receptor-interacting protein 1; MLKL, mixed lineage kinase domain-like; PI, propidium iodide.

To explore the necroptosis mechanism underlying the oncogenic role of TRAF5 in HCC, we assessed the protein expression levels of p-RIP1 (S166), RIP1, p-MLKL (S345), and MLKL in HCC cells. In this study, we found that sh-TRAF5 significantly up-regulated the protein expression ratios of p-RIP1 (S166)/RIP1 (P < 0.05) and p-MLKL (S345)/MLKL (P < 0.01) in HepG2 cells (Fig. 2G), indicating the pro-necroptosis effect of sh-TRAF5 on HepG2 cells. Besides, sh-TRAF5 notably decreased the survival ratio of HepG2 cells, but increased the necrosis and apoptotic ratios (P < 0.01; Fig. 2H). Results of Hoechst 33342/PI double-staining showed that sh-TRAF5 enhanced the number of necrotic and apoptotic HepG2 cells (Fig. 2I). Additionally, the detection results in HuH7 cells were comparable with those in HepG2 cells (Figs. S2G–S2I).

TRAF5 knockdown promotes the necroptosis of HCC cells

To further determine the necroptosis-promotive role of TRAF5 silencing on HCC cells, HepG2 and HuH7 cells were treated with sh-TRAF5 or/and the necroptosis inducer SHN. qRT-PCR and western blotting showed that SHN did not affect TRAF5 expression in HepG2 cells (P > 0.05; Figs. 3A and 3B). However, SHN exerted a notable inhibitory effect on HepG2 cell viability and a promotive effect on cell necrosis (P < 0.01; Figs. 3C and 3D). Moreover, SHN strengthened the ability of sh-TRAF5 to decrease the viability, inhibit the survival, and promote necrosis of HepG2 cells (P < 0.01; Figs. 3C and 3D), which were also confirmed by Hoechst 33342/PI double-staining assay (Fig. 3E). Furthermore, SHN enhanced the up-regulatory effect of sh-TRAF5I on the ratios of p-RIP1 (S166)/RIP1 (P < 0.01) and p-MLKL (S345)/MLKL (P < 0.05) in HepG2 cells (Fig. 3F). Similar results were observed in HuH7 cells (Figs. S3).

Figure 3 TRAF5 knockdown promotes the necroptosis of HCC cells.

(A and B) Detection of TRAF5 expression in HepG2 cells using qRT-PCR and western blotting. (C) Detection of the viability of HepG2 cells using CCK-8. (D) Detection of the survival, apoptotic, and necrosis ratios of HepG2 cells using flow cytometry. (E) Detection of the necrosis and apoptosis of HepG2 cells using Hoechst 33342/PI double-staining (Scale bar = 50 μm). (F) Detection of the protein expression of p-RIP1 (S166)/RIP1 and p-MLKL (S345)/MLKL in HepG2 cells using western blotting. Data were expressed as mean ± standard deviation. *P < 0.05 and **P < 0.01 vs vector group; #P < 0.05 and ##P < 0.01 vs sh-TRAF5 group; ns, no statistical significance between groups. TRAF5, TNF receptor-associated factor 5; HCC, hepatocellular carcinoma; qRT-PCR, quantitative real-time polymerase chain reaction; CCK-8, cell counting kit-8; RIP1 receptor-interacting protein 1; MLKL, mixed lineage kinase domain-like; PI, propidium iodide.

TRAF5 knockdown suppresses LTBR expression and NF- κB pathway

Research has suggested that the occurrence of HCC is related to LTBR signaling (Zhu et al., 2017). In the present study, down-regulation of LTBR expression (P < 0.01) induced by sh-TRAF5 was found in HepG2 cells with decreased TRAF5 expression (P < 0.01) (Fig. 4A). Besides, the interaction between TRAF5 and LTBR in HCC cells was determined using the co-IP assay (Fig. 4B). Immunofluorescence demonstrated that TRAF5 and LTBR co-localized in the cytoplasm of HEK293T cells (Fig. 4C).

Figure 4 TRAF5 positively regulates LTBR expression in HCC cells.

(A) Detection of TRAF5 and LTBR protein expression in HepG2 using western blotting. Data were expressed as mean ± standard deviation. **P < 0.01 vs sh-NC group. (B) Detection of the protein expression of TRAF5 and LTBR in HEK293T cells using co-IP. (C) Detection of the distribution of TRAF5 and HEK293T cells using immunofluorescence (Scale bar = 25 μm). TRAF5, TNF receptor-associated factor 5; LTBR, lymphotoxin beta receptor; HCC, hepatocellular carcinoma; co-IP, co-immunoprecipitation.

Subsequently, HepG2 and HuH7 cells were stably transfected with sh-LTBR-1 and sh-LTBR-2 to preliminarily investigate the involvement of LTBR in HCC development. Our data showed that sh-LTBR-1 and sh-LTBR-2 markedly down-regulated LTBR mRNA expression in both HepG2 and HuH7 cells relative to empty vector transfection (P < 0.01; Fig. S1C), indicating the successful knockdown of LTBR in HCC cells. Furthermore, the viability of HepG2 and HuH7 cells was significantly suppressed by sh-LTBR-1 and sh-LTBR-2 compared with empty vector transfection (P < 0.01; Fig. S1D). This suggested the beneficial role of sh-LTBR in preventing HCC cell malignant progression. To further validate the interaction between TRAF5 and LTBR in HCC, HepG2 cells were stably transfected with sh-TRAF5 or/and oe-LTBR. qRT-PCR and western blotting showed that oe-LTBR-transected HepG2 cells had markedly higher LTBR expression than control cells (P < 0.01; Figs. 5A and 5B), suggesting the successful transfection of HepG2 cells. Notably, oe-LTBR markedly abolished the effect of sh-TRAF5 on reducing LTBR expression in HepG2 cells (P < 0.01), whereas it exerted no significant impact on TRAF5 expression (P > 0.05; Figs. 5A and 5B). Meanwhile, oe-LTBR notably enhanced HepG2 cell viability (P < 0.01) and decreased the apoptotic ratio (P < 0.05) of HepG2 cells, which weakened the viability-suppressive effect (P < 0.05) and apoptosis-promotive (P < 0.01) effects of sh-TRAF5 (Figs. 5C and 5D). Moreover, oe-LTBR notably decreased the protein ratios of p-RIP1 (S166)/RIP1 (P < 0.01) and p-MLKL (S345)/MLKL (P < 0.01) in HepG2 cells (Fig. 5E). Similarly, oe-LTBR significantly eliminated the up-regulatory effect of sh-TRAF5 in the protein ratios of p-RIP1 (S166)/RIP1 (P < 0.01) and p-MLKL (S345)/MLKL (P < 0.05) in HepG2 cells (Fig. 5E).

Figure 5 TRAF5 knockdown blocks LTBR-medicated NF-κB pathway in HCC cells.

(A and B) Detection of the expression of TRAF5 and LTBR in HepG2 cells using qRT-PCR and western blotting. (C) Detection of the viability of HepG2 cells using CCK-8. (D) Detection of the apoptosis of HepG2 cells using flow cytometry. (E) Detection of the expression of p-RIP1 (S166)/RIP1, p-MLKL (S345)/MLKL, and p-NF-κB/NF-κB using western blotting. Data were expressed as mean ± standard deviation. *P < 0.05 and **P < 0.01 vs vector group; #P < 0.05 and ##P < 0.01 vs sh-TRAF5 group; ns, no statistical significance between groups. TRAF5, TNF receptor-associated factor 5; LTBR, lymphotoxin beta receptor; NF-κB, nuclear factor kappaB; HCC, hepatocellular carcinoma; qRT-PCR, quantitative real-time polymerase chain reaction; CCK-8, cell counting kit-8; RIP1 receptor-interacting protein 1; MLKL, mixed lineage kinase domain-like.

Evidence has demonstrated that the activation of NF-κB signaling can promote the malignant progression of HCC (Ding et al., 2020). Thus, we also assessed the levels of NF-κB pathway-related proteins (p-NF-κB and NF-κB). Here, we observed that sh-TRAF5 significantly down-regulated the p-NF-κB/NF-κB protein ratio in HepG2 cells (P < 0.01), while oe-LTBR markedly up-regulated this ratio (P < 0.01; Fig. 5E). Notably, oe-LTBR abolished the effect of sh-TRAF5 on decreasing the p-NF-κB/NF-κB protein ratio in HepG2 cells (P < 0.01; Fig. 5E).

TRAF5 knockdown elevates the necroptosis of HCC cells by blocking the NF- κB pathway

To further ascertain whether TRAF5 knockdown could inhibit the NF-κB signaling in HCC cells, HepG2 cells were stably transfected with sh-NC or sh-TRAF5 and the NF-κB pathway was further analyzed. Western blotting showed that sh-TRAF5 notably down-regulated the p-NF-κB/NF-κB protein level in HepG2 cells (P < 0.01; Fig. 6A). Also, to further verify whether TRAF5 silencing could enhance HCC cell necroptosis by blocking NF-κB signaling, the NF-κB inhibitor SC75741 was administered to sh-TRAF5-transfected HepG2 cells. Herein, we found that SC75741 significantly down-regulated both NF-κB mRNA level (P < 0.01) and p-NF-κB/NF-κB protein ratio (P < 0.01) in HepG2 cells compared with those in control cells, without affecting TRAF5 expression (P > 0.05; Figs. 6B and 6C). Moreover, SC75741 enhanced the ability of sh-TRAF5 to suppress the NF-κB pathway (P < 0.01) and elevate the protein ratios of p-RIP1 (S166)/RIP1 (P < 0.01) and p-MLKL (S345)/MLKL (P < 0.01) in HepG2 cells (Figs. 6B and 6C).

Figure 6 TRAF5 knockdown enhances the necroptosis of HCC cells by blocking NF-κB pathway.

(A) Detection of the protein expression level of p-NF-κB/NF-κB in HepG2 cells using western blotting. (B) Detection of NF-κB mRNA expression in HepG2 cells using qRT-PCR. (C) Detection of protein expression of TRAF5, p-NF-κB/NF-κB, p-RIP1 (S166)/RIP1, and p-MLKL (S345)/MLKL in HepG2 cells using western blotting. Data were expressed as mean ± standard deviation. *P < 0.05 and **P < 0.01 vs sh-NC group; ##P < 0.01 vs sh-TRAF5 group; ns denotes no statistical significance between groups. TRAF5, TNF receptor-associated factor 5; HCC, hepatocellular carcinoma; NF-κB, nuclear factor kappaB; qRT-PCR, quantitative real-time polymerase chain reaction; RIP1 receptor-interacting protein 1; MLKL, mixed lineage kinase domain-like.

TRAF5 knockdown inhibits the malignant progression of HCC in vivo

To further figure out whether TRAF5 silencing could suppress HCC tumor progression in vivo, a xenograft model was constructed by administering sh-TRAF5-transfected HepG2 cells to mice. We found that sh-TRAF5 markedly reduced the volume (P < 0.01) and weight (P < 0.01) of xenograft tumors (Fig. 7A). Immunohistochemistry demonstrated the decreased expression of Ki67 (a cell proliferation marker) in xenograft tumor tissues by sh-TRAF5 (Fig. 7B). Besides, sh-TRAF5 promoted cell apoptosis in xenograft tumor tissues (Fig. 7C). Meanwhile, the protein ratios of p-RIP1 (S166)/RIP1 (P < 0.01) and p-MLKL (S345)/MLKL (P < 0.01) in xenograft tumor tissues were markedly increased by sh-TRAF5 (Fig. 7D).

Figure 7 TRAF5 knockdown suppresses the malignant progression of HCC in vivo.

(A) Detection of the volumes and weights of tumors (n = 6). (B) Detection of Ki67 protein expression to determine cell proliferation in xenograft tumor tissues using immunohistochemistry (Scale bar = 20 μm). (C) Detection of the cell apoptosis in xenograft tumor tissues using TUNEL staining (Scale bar = 20 μm). (D) Detection of the protein expression of p-RIP1 (S166)/RIP1 and p-MLKL (S345)/MLKL using western blotting (n = 6). Data were expressed as mean ± standard deviation. **P < 0.01 vs sh-NC group. TRAF5, TNF receptor-associated factor 5; HCC, hepatocellular carcinoma; TUNEL, terminal deoxynucleotidyl transferase-mediated dUTP nick-end labeling; RIP1 receptor-interacting protein 1; MLKL, mixed lineage kinase domain-like.

Discussion

HCC is an aggressive malignant tumor with high mortality (Chedid et al., 2017). TRAF5 has been determined as an oncogene in some cancers, such as melanoma (Ma, Duan & Hao, 2020) and ovarian cancer (Zhang et al., 2022). Interestingly, a recent study showed that inhibition of TRAF5 expression can suppress the malignant progression of HCC cells (Ding et al., 2021a). However, the specific role of TRAF5 in the onset and development of HCC remains largely unclear. These facts prompted us to further investigate the oncogenic mechanisms of TRAF5 in HCC. This study confirmed the up-regulated expression of TRAF5 in HCC cell lines, and TRAF5 knockdown suppressed HCC cell proliferation, migration, and invasion, as well as promoted HCC cell apoptosis. Besides, in vitro experiments showed that TRAF5 regulated proliferation, apoptosis, and necroptosis of HCC cells by targeting molecules, NF-κB, RIPIs, and MLKLs. Importantly, our results showed that TRAF5 knockdown (sh-TRAF5) suppressed the viability, colony formation, migration, and invasion of HCC cells. Collectively, this study revealed that TRAF5 promoted the malignant phenotypes and progression of HCC by regulating the LTBR/NF-κB and RIPI/MLKL signaling pathways.

Accumulating studies have demonstrated that TRAF5 is implicated in tumor cell proliferation, apoptosis, and metastasis (Li et al., 2016; Liang et al., 2019). Further, an investigation has indicated the necroptosis-inhibitory role of TRAF5 in colorectal cancer (Wu et al., 2021). In line with the previous study (Wu et al., 2021), our data showed that TRAF5 deficiency promoted HCC cell necroptosis, as the ratios of p-RIP1 (S166)/RIP1 and p-MLKL (S345)/MLKL were increased. Meanwhile, TRAF5 knockdown inhibited the survival and enhanced apoptosis of HCC cells, which was consistent with previous data (Ding et al., 2021b). These results indicated the involvement of TRAF5 in HCC development. On the other hand, evidence shows that TRAF5 is capable of directly binding to the cytoplasmic portion of LTBR and correlating with LTBR in immune cell migration (Piao et al., 2021). LTBR plays a crucial role in regulating lymphoid organ development and enhancing immune and inflammatory responses (Das et al., 2019). LTBR is not only expressed in lymphoid cells but also in a variety of tumor types, including numerous solid tumors (Das et al., 2019). Importantly, a study has indicated the association of LTBR polymorphisms in the onset of HCC (Zhu et al., 2017). Based on these previous studies, we hypothesized an interaction between TRAF5 and LTBR in HCC. Interestingly, our results showed that TRAF5 knockdown significantly decreased LTBR expression in HCC cells. Besides, the results of co-IP and immunofluorescence further indicated an inner link between TRAF5 and LTBR in HCC cells. A previous study found that LTBR was increased in head and neck cancer, which could induce tumor cell migration (Das et al., 2019). Moreover, LTBR has been indicated to trigger the proliferation of colorectal cancer cells (Buhrmann et al., 2019). Significantly, LTBR signaling was demonstrated to promote cell proliferation, which could conduce to the development of liver tumors (Haybaeck et al., 2009). Consistent with the previous findings (Haybaeck et al., 2009), our data showed that LTBR knockdown effectively inhibited HCC cell proliferation. Furthermore, we observed that LTBR overexpression reversed TRAF5 deficiency-induced decrease in LTBR expression and cell viability, as well as the increase in apoptosis and necroptosis in HCC cells. Collectively, these findings indicate that TRAF5 knockdown can promote necroptosis by down-regulating LTBR expression in HCC cells.

To further investigate the mechanisms by which TRAF5 knockdown reduces LTBR expression to enhance HCC cell necroptosis, the signaling pathway was subsequently explored in this study. Among various oncogenic pathways, the role of NF-κB signaling in tumor metastasis as a core inflammation-regulatory transcription factor has long been a hot topic (Ahmad et al., 2022). The NF-κB signaling cascade has implications for multiple cellular processes, including cell proliferation and apoptosis (Zhou et al., 2020). Over-activation of NF-κB signaling is reportedly associated with the pathogenesis of various cancers, including HCC (Dai et al., 2020; Li et al., 2019; Wei et al., 2020). Importantly, NF-κB can be a direct target of TRAF5 to protect from tumor cell death in some malignancies, such as gastric cancer (Xie et al., 2019) and colon cancer (Sun et al., 2020). TRAF5 can directly bind to a small region in the cytoplasmic portion of LTBR to initiate NF-κB activation, which leads to the expression of pro-inflammatory effectors (Piao et al., 2021). In the current study, we found that the NF-κB pathway in HCC cells was suppressed following TRAF5 knockdown, however, which was abolished by LTBR overexpression. In line with the present study, a previous study demonstrated that LTBR could induce the activation of the NF-κB pathway in head and neck cancer (Yang et al., 2019). Moreover, we observed that the NF-κB inhibitor SC75741 significantly enhanced the pro-necroptosis effect of TRAF5 knockdown in HCC cells. Previous research revealed that LTBR could stimulate NF-κB-dependent transcription and facilitates apoptosis (Banach-Orłowska et al., 2019). Furthermore, another study revealed that activated LTBR can promote tumor cell migration and metastasis by activating the NF-κB signaling cascade in head and neck cancer (Das et al., 2019). Taken together, these data indicate that TRAF5 silencing can accelerate HCC cell necroptosis by blocking LTBR-mediated NF-κB signaling pathway.

To further verify the beneficial role of TRAF5 silencing in preventing HCC progression revealed above, an HCC xenograft model was established and in vivo assays were conducted. Our data showed that down-regulation of TRAF5 inhibited xenograft tumor growth, as the volume and weight of xenograft tumors were decreased following TRAF5 deficiency. Additionally, Ki67 protein expression (a signature of cell proliferation) in xenograft tumor tissues was reduced after TRAF5 knockdown. Meanwhile, TRAF5 deficiency enhanced cell necroptosis in xenograft tumor tissues as the number of apoptotic cells and protein expression levels of p-RIP1 (S166)/RIP1 and p-MLKL (S345)/MLKL were increased. Collectively, these findings imply that down-regulation of TRAF5 can restrain tumor growth, suppress cell proliferation, and promote cell necroptosis in HCC.

Our study had some limitations. There are multiple molecules and mechanisms, such as inflammatory responses and related cytokines, involved in the LTBR signaling in the development of HCC. However, we merely focused on its downstream target, NF-κB, and the necroptosis mechanism. In addition, we validated the inhibitory effect of TRAF5 knockdown on HCC cell necroptosis in vivo, but without in vivo verification of the NF-κB pathway. Therefore, the carcinogenic mechanisms of TRAF5 in HCC require to be further explored.

Conclusions

Our research demonstrated that TRAF5 silencing could promote necroptosis by inhibiting LTBR-mediated NF-κB pathway to delay the malignant progression of HCC. This study revealed the mechanisms underlying the carcinogenic effect of TRAF5 in HCC, which furthers our insights into the mechanisms of HCC and provides novel therapeutic strategies for HCC.

Supplemental Information

Supplemental Information 1 All immunoblot replicates.

Click here for additional data file.

Supplemental Information 2 TRAF5 and LTBR knockdown inhibit HCC cell proliferation.

(A) Detection of TRAF5 mRNA expression in HepG2 and HuH7 using qRT-PCR. (B) Detection of the viability of HepG2 and HuH7 cells using CCK-8. (C) Detection of LTBR mRNA expression in HepG2 and HuH7 using qRT-PCR. (D) Detection of the viability of HepG2 and HuH7 cells using CCK-8. Data were expressed as mean ± standard deviation. **P < 0.01 vs vector group. TRAF5, TNF receptor-associated factor 5; LTBR, lymphotoxin beta receptor; HCC, hepatocellular carcinoma; qRT-PCR, quantitative real-time polymerase chain reaction; CCK-8, cell counting kit-8.

Click here for additional data file.

Supplemental Information 3 TRAF5 knockdown suppresses the malignant progression of HCC cells.

(A and B) Detection of TRAF5 expression in HuH7 cells using qRT-PCR and western blotting. (C) Detection of the viability of HuH7 cells using CCK-8. (D) Detection of the colony formation of HuH7 cells. (E) Detection of the migration of HuH7 using cell wound healing assay (Scale bar = 50 μm). (F) Detection of the invasion of HuH7 cells using Transwell assay (Scale bar = 50 μm). (G) Detection of the protein expression of p-RIP1 (S166)/RIP1 and p-MLKL (S345)/MLKL in HuH7 cells using western blotting. (H) Detection of the survival, apoptotic, and necrosis ratios of HuH7 cells using flow cytometry. (I) Detection of the necrosis and apoptosis of HuH7 cells using Hoechst 33342/PI double-staining. Scale bar = 50 μm. Data were expressed as mean ± standard deviation. **P < 0.01 vs sh-NC group. TRAF5, TNF receptor-associated factor 5; HCC, hepatocellular carcinoma; qRT-PCR, quantitative real-time polymerase chain reaction; CCK-8, cell counting kit-8; RIP1 receptor-interacting protein 1; MLKL, mixed lineage kinase domain-like; PI, propidium iodide.

Click here for additional data file.

Supplemental Information 4 TRAF5 knockdown enhances the necroptosis of HCC cells.

(A and B) Detection of TRAF5 expression in HuH7 cells using qRT-PCR and western blotting. (C) Detection of the viability of HuH7 cells using CCK-8. (D) Detection of the survival, apoptotic, and necrosis ratios of HuH7 cells using flow cytometry. (E) Detection of the necrosis and apoptosis of HuH7 cells using Hoechst 33342/PI double-staining. Scale bar = 50 μm. (F) Detection of the protein expression of p-RIP1 (S166)/RIP1 and p-MLKL (S345)/MLKL in HuH7 cells using western blotting. Data were expressed as mean ± standard deviation. **P < 0.01 vs vector group; ##P < 0.01 vs sh-TRAF5 group; ns denotes no statistical significance between groups. TRAF5, TNF receptor-associated factor 5; HCC, hepatocellular carcinoma; qRT-PCR, quantitative real-time polymerase chain reaction; CCK-8, cell counting kit-8; RIP1 receptor-interacting protein 1; MLKL, mixed lineage kinase domain-like; PI, propidium iodide.

Click here for additional data file.

Supplemental Information 5 ARRIVE checklist.

Click here for additional data file.

Additional Information and Declarations

Competing Interests

Author Contributions

Animal Ethics

Data Availability

The authors declare that they have no competing interests.

Guolin Wu conceived and designed the experiments, performed the experiments, analyzed the data, prepared figures and/or tables, authored or reviewed drafts of the article, and approved the final draft.

Fangping Wu conceived and designed the experiments, performed the experiments, analyzed the data, prepared figures and/or tables, authored or reviewed drafts of the article, and approved the final draft.

Yang Qing Zhou conceived and designed the experiments, performed the experiments, authored or reviewed drafts of the article, and approved the final draft.

Wenwen Lu performed the experiments, analyzed the data, authored or reviewed drafts of the article, and approved the final draft.

Feng Lin Hu performed the experiments, analyzed the data, authored or reviewed drafts of the article, and approved the final draft.

Xiaofen Fan performed the experiments, analyzed the data, authored or reviewed drafts of the article, and approved the final draft.

The following information was supplied relating to ethical approvals (i.e., approving body and any reference numbers):

All animal experiments were approved by the Animal Ethics Committee of Xiamen University.

The following information was supplied regarding data availability:

Guolin Wu. (2023). Silencing of TRAF5 enhances necroptosis in hepatocellular carcinoma by inhibiting LTBR-mediated NF-κB signaling [Data set]. Zenodo. https://doi.org/10.5281/zenodo.7690290.

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
