# Peer review of "Silencing of TRAF5 enhances necroptosis in hepatocellular carcinoma by inhibiting LTBR-mediated NF-κB signaling"

_PeerJ, doi:10.7717/peerj.15551_

## Round 0.1 · original submission · Major Revisions

The author should answer these Reviewers' questions carefully.

Reviewer 1 ·

Basic reporting

In the current manuscript Wu et al presented the results of their study of the molecular mechanism of TRAF5 cell death regulation in hepatocellular carcinoma cell (HCC) lines. TRAF5-based LTBR-mediated NF-kB signaling was proposed to be important for viability of the tested cell lines. Unfortunately, the results presented in the manuscript do not support the authors conclusion. To the great extent it is due to the problem with setting the appropriate experimental design. Additionally, manuscript may gain clarity upon improvement in English writing.

Experimental design

Major points:
1. To test the signaling activity of TRAF5 in HCC lines, authors transfected cells with sh-TRAF5 vectors. It is not clear if the authors established stable or transient cell lines. If the stable cell lines were used in the study, then authors should establish and test several lines with downregulated TRAF5 expression to avoid clonal variation. In this case authors should characterize established cell lines such as expression of the molecules partaking in the signaling activities of LTBR (NF-kB, MAPK activation apoptotic and necroptotic cell death induction). If cell lines used express sh-TARF5 transiently, then authors should test TRAF5 expression levels in the tumors (Fig. 7) throughout the study. In any case, downregulation of TRAF5 protein expression levels achieved was partial. I would recommend CRISPR technology to generate TRAF5-null cell lines.
2. The cell death data presented in the figures 2 and S1, was it spontaneous cell death without any stimulation? What is the mechanism? If LTBR signaling was suggested authors should generate LTBR-deficient cell lines and test viability of those cells.
3. Is TRAF5 partake in LTBR protein signaling complex in HCC lines studied? How do these cells respond to LTB treatment?
Minor points:
1. Mw markers should be indicated in the figures containing WB images.
2. The legends in the figures 2a, b, d -g; 3a-d; 5a-d; 6b, 7d, s1a, b, d-g, s2a-d are missing

Validity of the findings

The results presented in the manuscript do not support the authors conclusion.

·

Basic reporting

In the current study, Wu and colleagues explored the role of TRAF5 in the development of HCC in in vitro cultured cells and in vivo xenograft model. They found that knockdown of TRAF5 down-regulated LTBR and promoted apoptosis as well as necroptosis possibly through inhibition of NF-κB pathway. Overall, these are the novel observations, and the data support the main conclusion. However, there are minor concerns that need to be addressed, which are listed as follows:
1. It is better that the authors can add the molecular weight of the WB blots in the figures.
2. Please check the blot of Figure 5B is correct or not correct. In this blot, I cannot see the effects of sh-TRAF5 or oe-LTBR.
3. Some grammar or typo errors have been identified. The authors should check the manuscript carefully or seek help from certified English Editing Service to polish its English. For example, in line 98, (' et al., 2019).

Experimental design

Yes

Validity of the findings

Yes

Reviewer 3 ·

Basic reporting

The present article "Silencing of TRAF5 enhances necroptosis in hepatocellular carcinoma by inhibiting LTBR mediated NF-κB signaling” is interesting and provided information about TRAF5. The authors analyzed the pathogenic mechanism of TRAF5 in HCC both in vivo and in vitro, which could provide new therapeutic strategies for the treatment of HCC. The study is well-designed and executed. However, the authors should address the following concerns.
1. Results description needs extensive revision. In some cases, it would be important to add the numeric description of the data to highlight the significance. For instance, a 10% increase is not the same as a 50%.
2、The name of the cell line should be addressed clearly in the figures.
3、Discussion could be written in a more compact way. It stands as each paragraph referring to distinct subjects. Please merge the data with the literature and clearly show the gaps that were filled with this paper.

Experimental design

no comment

Validity of the findings

no comment

Additional comments

no comment

---

## Round 0.2 · Minor Revisions

The review 1 should be better addressed. Also, the authors should upload all replicates of the immunoblots. They need also to clarify how many technical and biological replicates were performed for all experiments.

Reviewer 1 ·

Basic reporting

In my original review I pointed to authors on three areas of their work that require significant improvements. Unfortunately, authors did not manage to provide required data.
In particular:
I asked:
1. To test the signaling activity of TRAF5 in HCC lines, authors transfected cells with sh-TRAF5 vectors. It is not clear if the authors established stable or transient cell lines. If the stable cell lines were used in the study, then authors should establish and test several lines with downregulated TRAF5 expression to avoid clonal variation. In this case authors should characterize established cell lines such as expression of the molecules partaking in the signaling activities of LTBR (NF-kB, MAPK activation apoptotic and necroptotic cell death induction). If cell lines used express sh-TARF5 transiently, then authors should test TRAF5 expression levels in the tumors (Fig. 7) throughout the study. In any case, downregulation of TRAF5 protein expression levels achieved was partial. I would recommend CRISPR technology to generate TRAF5-null cell lines.

Authors established stable cell lines, but they did not respond to my request:” If the stable cell lines were used in the study, then authors should establish and test several lines with downregulated TRAF5 expression to avoid clonal variation. In this case authors should characterize established cell lines such as expression of the molecules partaking in the signaling activities of LTBR (NF-kB, MAPK activation apoptotic and necroptotic cell death induction).”

2. The cell death data presented in the figures 2 and S1, was it spontaneous cell death without any stimulation? What is the mechanism? If LTBR signaling was suggested authors should generate LTBR-deficient cell lines and test viability of those cells.

Authors did not answer my questions, but provided irrelevant answers.

3. Is TRAF5 partake in LTBR protein signaling complex in HCC lines studied? How do these cells respond to LTB treatment?

Authors did not answer my questions, but provided irrelevant answers.

Experimental design

Authors did not answer my questions in full.

Validity of the findings

Authors did not answer my questions in full.

·

Basic reporting

The authors have addressed my concerns.

Experimental design

The authors have addressed my concerns.

Validity of the findings

The authors have addressed my concerns.

Reviewer 3 ·

Basic reporting

no comment

Experimental design

no comment

Validity of the findings

no comment

---

## Round 0.3 · accepted · Accept

After carefully checking, this is a good revised manuscript.